# Indoor Positioning System Based on Global Positioning System Signals with Down- and Up-Converters in 433 MHz ISM Band

**DOI:** 10.3390/s21134338

**Published:** 2021-06-25

**Authors:** Abdulkadir Uzun, Firas Abdul Ghani, Amir Mohsen Ahmadi Najafabadi, Hüsnü Yenigün, İbrahim Tekin

**Affiliations:** 1Electronics Engineering, Sabanci University, Istanbul 34956, Turkey; kadiruzun@sabanciuniv.edu (A.U.); firas@sabanciuniv.edu (F.A.G.); amirahmadi@sabanciuniv.edu (A.M.A.N.); 2Radar Electronic Warfare and Intelligence Systems Division, ASELSAN A.Ş., Istanbul 34906, Turkey; 3Computer Science and Engineering, Sabanci University, Istanbul 34956, Turkey; yenigun@sabanciuniv.edu

**Keywords:** down-conversion, GPS, indoor positioning, navigation, RF repeaters, up-conversion

## Abstract

In this paper, an indoor positioning system using Global Positioning System (GPS) signals in the 433 MHz Industrial Scientific Medical (ISM) band is proposed, and an experimental demonstration of how the proposed system operates under both line-of-sight and non-line-of-sight conditions on a building floor is presented. The proposed method is based on down-converting (DC) repeaters and an up-converting (UC) receiver. The down-conversion is deployed to avoid the restrictions on the use of Global Navigation Satellite Systems (GNSS) repeaters, to achieve higher output power, and to expose the GPS signals to lower path loss. The repeaters receive outdoor GPS signals at 1575.42 MHz (L1 band), down-convert them to the 433 MHz ISM band, then amplify and retransmit them to the indoor environment. The front end up-converter is combined with an off-the-shelf GPS receiver. When GPS signals at 433 MHz are received by the up-converting receiver, it then amplifies and up-converts these signals back to the L1 frequency. Subsequently, the off-the-shelf GPS receiver calculates the pseudo-ranges. The raw data are then sent from the receiver over a 2.4 GHz Wi-Fi link to a remote computer for data processing and indoor position estimation. Each repeater also has an attenuator to adjust its amplification level so that each repeater transmits almost equal signal levels in order to prevent jamming of the off-the-shelf GPS receiver. Experimental results demonstrate that the indoor position of a receiver can be found with sub-meter accuracy under both line-of-sight and non-line-of-sight conditions. The estimated position was found to be 54 and 98 cm away from the real position, while the 50% circular error probable (CEP) of the collected samples showed a radius of 3.3 and 4 m, respectively, for line-of-sight and non-line-of-sight cases.

## 1. Introduction

Global indoor positioning is an emerging market whose size is forecast to grow from USD 6.1 billion to USD 17.0 billion by 2025 [1]. Many researchers, in the field of indoor positioning, have proposed different solutions to solve this sophisticated problem. Some of the proposed technologies for indoor positioning are based on IEEE 802.11 [2,3,4,5,6,7], Bluetooth [8,9,10,11], Zigbee [12], radio frequency identification devices (RFIDs) [13], visible light [14], acoustic [15,16], and ultrasound [17,18]. GNSS-based solutions have also proven to be a candidate for the indoor positioning problem [19,20,21]. Some of the GNSS-based solutions are based on high sensitivity GNSS (HS-GNSS) [22], assisted GNSS (A-GNSS) [23], pseudolites [24,25,26,27,28,29,30,31], GNSS repeaters [32,33], repealites [34,35], and peer-to-peer cooperative positioning [36]. Among the GNSS-based techniques, HS-GNSS and A-GNSS technologies require no infrastructure within the indoor environment, while pseudolite and repeater-based approaches require infrastructure.

Ideally, from a navigational point of view, a GPS-based solution to the indoor positioning problem would result in integrated outdoor and indoor applications such as asset tracking, vehicular navigation, and emergency services. However, GPS-based solutions currently experience a multitude of weaknesses due to (a) reflections and multipath errors from the indoor environment, (b) attenuation (up to 30 dB [37]) from walls and other structural components, (c) non-line-of-sight conditions caused by corners and objects, (d) changes in the indoor environment due to people and moving objects, and (e) RF devices operating in the same frequency band and their interference to the indoor positioning system.

In addition to the listed drawbacks, GPS-based systems operating in L1, L2, and L5 bands also suffer from the restrictions on the use of GNSS repeaters. Such restrictions aim to prevent repeaters from interfering with other GNSS systems in the vicinity. The Electronic Communications Committee (ECC), European Telecommunication Standards Institute (ETSI), and the US policy “Manual of Regulations and Procedures for Federal Radio Frequency Management” present the practices and restrictions on the use of GNSS repeaters [38,39,40,41]. These standards reduce the coverage of GPS repeaters by limiting the output power and the maximum allowable amplification for repeater structures in the L1, L2, and L5 frequency bands.

In [32,42], a GPS-based indoor positioning technique with three repeaters is shown; however, the repeaters operate solely in the L1 band and, therefore, are restricted to the aforementioned restrictive policies in their usage.

This paper presents a new GPS-based approach that does not contradict the restrictions in the aforementioned standards. The proposed indoor positioning system in this paper operates in the 433 MHz ISM band, hence it is not subject to the amplification restrictions in GPS frequencies. The proposed repeaters down-convert GPS signals in the 1575.42 MHz (L1 band) to the 433 MHz ISM band, allowing signal coverage to be increased, with the higher permitted power levels in the 433 MHZ ISM band. In addition to the higher power levels permitted in the 433 MHz ISM band, the free space path loss in the 433 MHz frequency is 11.22 dB less, as presented in Figure 1, while the penetration through walls is higher than at 1575.42 MHz (GPS frequency) or 2.4 GHz (Wi-Fi frequency). Therefore, in terms of coverage and compatibility with existing rules on the use of repeaters, the proposed system could outperform the existing indoor positioning systems that are operating at frequencies higher than 433 MHz.

The system we propose in this paper differs from the existing systems with its following properties. The down- and up-conversion schemes have previously been proposed in [43,44,45] for indoor positioning applications with GPS signals. While in [43,44], a down-converting repeater is proposed, neither indoor positioning nor GPS signal down-conversion is demonstrated. In [45], only the down-converting repeater and up-converting receiver circuits are presented, to show that GPS signal fidelity is preserved. However, in [45], it is not shown how the proposed circuits perform in an indoor environment for positioning. In [46], the indoor positioning of an up-converting receiver at a point on a line between two down-converting repeaters (1-dimensional (1D) positioning) is proposed; however, non-line-of-sight conditions and two-dimensional (2D) positioning of the receiver are not addressed. In [47], we refer to the patent application of a system that can be used with different positioning systems (i.e., BeiDou, Galileo, GLONASS) by changing the register sets in the down- and up-converters to work in the 433 MHz frequency band or even in other ISM bands. In the latter case, the indoor antennas should also be changed to the selected frequency band.

Upon review of readily available publications and to the best of the authors’ knowledge, this paper is the first experimental study that demonstrates that 2D indoor positioning can be achieved by transmitting GPS signals, at 433 MHz, from three repeaters in an indoor environment, where the position of the receiver can be estimated by calculating the distance between each repeater and the receiver, on the plane that is formed by the deployed repeaters. The indoor positioning concept that is proposed in this paper is described visually in Figure 2.

In this particular paper, we present two experiments for 2D indoor positioning in the 433 MHz ISM band: one experiment is under line-of-sight conditions, and another is under non-line-of-sight conditions. Note that adding a 4th repeater will allow us to achieve 3D indoor positioning.

The rest of the paper is organized as follows: Section 2 introduces the proposed indoor positioning system and describes its hardware, software and algorithms. Section 3 describes the experimental framework and the real-life environment where we tested our system. Section 4 analyzes and discusses the results we achieved when the proposed system and technique are used for 2D indoor positioning. Finally, Section 5 concludes this paper.

## 2. Indoor Positioning System in 433 MHz ISM Band

The proposed indoor positioning system is composed of two unique subsystems: GPS down-converting repeaters and an up-converter integrated with an off-the-shelf GPS receiver. The block diagrams of the GPS down-converting repeaters and the up-converter with the off-the-shelf GPS receiver are presented in Figure 3. The red dashed line represents the 433 MHz RF link.

### 2.1. GPS Down-Converting Repeater

The GPS down-converting repeater subsystem comprises RF blocks and supporting blocks, which are presented in Table 1. The RF blocks refer to the components that together form the RF path, through which the GPS signal propagates, while the supporting blocks sustain the regular operation of RF blocks. The GPS down-converting repeater subsystem is implemented using the listed components and blocks in Table 1.

The implemented GPS down-converting repeater is depicted in Figure 4. Fabricated and modelled 1575.42 MHz outdoor directional GPS antenna and 433 MHz dipole antenna are demonstrated in Figure 5. The RF blocks of the GPS down-converting repeater subsystem are as follows; a directional outdoor GPS antenna (proposed in [32]) and its bias-tee, a 1575.42 to 433 MHz down-converter, a signal power conditioner and filter block (where an LNA, a variable attenuator, a 433 MHz band pass filter, and a second LNA are cascaded), and an indoor 433 MHz dipole antenna. The supporting blocks are a controller over Wi-Fi that interfaces between the RF blocks and the user, and a voltage regulator that provides the required DC to RF blocks.

The active directional GPS antennas pick-up the GPS signals from satellites in the 3 dB beam width. The conic reflector reduces the beam width by 30 degrees. Therefore, the 3 dB beam width of the fabricated antenna is measured as 60 degrees. A bias-tee is placed before the down-converter to provide the DC voltage that is required by the active directional GPS antenna.

When the GPS signal at 1575.42 MHz is received by the antenna, the signal passes to the down-converter. The down-converter converts the signal from 1575.42 to 433 MHz. The amplifiers in the signal power conditioner and filter block then amplify the signal. The amplification level can be adjusted by changing the attenuation of the digital step attenuator, which is able to attenuate the signal from 0 to 31.5 dB. By adjusting the attenuation value of each attenuator, the gain of the overall down-converter subsystem may be set to different values. The bandpass filter at 433 MHz is also deployed to eliminate signals and harmonics out of the band. Furthermore, the signal power conditioner block has another amplifier following the band pass filter. After the second amplifier, the down-converted GPS signals are retransmitted to indoors via a 433 MHz dipole antenna at each repeater.

In terms of adjustments, the repeater board attenuation levels can be set to different values from a remote computer using a Wi-Fi connection. In this way, the gain of a repeater can be adjusted to prevent near-far effects and also keep the signal level from each repeater at a similar level. With this structure, the down-conversion of GPS signals enables users to deploy higher gain GPS receivers than those that are limited by international standards. Additionally, operating in 433 MHz allows us to further increase the gain of the proposed repeaters.

### 2.2. Up-Converting Receiver

The up-converting receiver subsystem is comprised of the RF blocks and supporting blocks, which are presented in Table 2.

The up-converting receiver is implemented using the listed components and blocks in Table 2 and is presented in Figure 6. The aforementioned 433 MHz dipole antenna in Figure 5e is also used as the receiver indoor antenna. The RF blocks of the up-converting receiver are (a) a 433 MHz indoor dipole antenna, (b) a signal power conditioner block where a 433 MHz band pass filter, (c) an LNA, (d) a variable attenuator, (e) a second, cascaded LNA, (f) an I/Q power divider, (g) bias tees that provide the required DC to I/Q inputs of the up-converter, (h) a 433 to 1575.42 MHz up-converter, and (i) an off-the-shelf GPS receiver (u-Blox LEA-6T^®^). The controller over Wi-Fi provides interfaces between RF blocks and the user. Moreover, a voltage regulator is designed to provide the required DC voltage to the RF blocks. In this subsystem, the controller is also connected to the custom GPS receiver, and the raw data from the GPS receiver can be sent over Wi-Fi to a remote computer.

The retransmitted 433 MHz positioning signals are picked up by the 433 MHz indoor dipole antennas. In the receiver subsystem, the received signals are first filtered and then amplified. Similar to the repeater subsystem, the amplification levels are adjusted by changing the attenuation of the digital step attenuator, which is able to attenuate the signal from 0 to 31.5 dB. By adjusting the attenuation value of the attenuator, the gain of the overall up-converter subsystem may be set to different values. The subsystem also utilizes a second amplifier to further amplify the signal. These up-converted GPS signals then propagate to the off-the-shelf GPS receiver, which is integrated within the up-converting receiver subsystem.

As mentioned previously, for further flexibility, the up-converting repeater attenuation levels can be set to different values from a remote computer using a Wi-Fi connection. Using a remote computer with an internet connection, where the ephemeris data of GPS satellites are available, the estimation of the indoor position starts when the raw data are sent from the receiver. Therefore, one can conclude that the established Wi-Fi link between the proposed up-converting receiver, and the computer where calculations are done will provide a hot start to the system.

### 2.3. Algorithm for Indoor Position Estimation

In an indoor environment, although the signal loss and GPS coverage problems can be overcome with the proposed repeaters, this solution requires additional algorithms that takes the non-line-of-sight propagation and repeater delay into account. The proposed technique introduces a new path in that the distance between satellite and the receiver becomes different from that of the normal operation of an off-the-shelf receiver during which there is a line-of-sight distance between the satellite and the receiver. The GPS signals in the proposed scheme come to the repeater first and, then, reach to the receiver as seen in Figure 2.

As part of the indoor position estimation, an algorithm (run on MATLAB^®^) is designed and run on a remote computer that is connected to the system via Wi-Fi. The Wi-Fi link is established with a Wi-Fi module on the controller block of the receiver board. The raw data (such as pseudo-range, carrier-to-noise ratio, satellite azimuth, and elevation angles, etc.) obtained from the off-the-shelf receiver are transmitted to the remote computer through the Wi-Fi connection. The routine summarized in Figure 7 has the following four steps: raw data reception, satellite selection and satellite-to-repeater distance calculations, cleaning pseudo-range from satellite-to-repeater distances and satellite biases, and finally, running the least squares navigation (LSNAV) algorithm, which is a least squares solution that minimizes the sum of the square of the residual errors [48].

In the first part, we remove the biases from the GPS pseudo-range measurement using the models of the troposphere, ionosphere, GPS satellite clocks, GPS satellite movement during signal propagation, and Earth rotation.

Navigation data are also collected as part of raw data from the off-the-shelf receiver, which contains GPS time of the week (ITOW) of the navigation epoch. ITOW field indicates the GPS time at which the navigation epoch occurred. Each navigation solution is triggered by the tick of the 1 kHz clock nearest to the desired navigation solution time. This tick is referred to as a navigation epoch. If the navigation solution attempt is successful, one of the results is an accurate measurement of time in the time base of the chosen GNSS system, called GNSS system time. The difference between the calculated GNSS system time and receiver local time is called the clock bias (and the clock drift is the rate at which this bias is changing). Navigation data also contain an Earth-centered Earth-fixed (ECEF) coordinates solution of the off-the-shelf receiver; however, this solution would not be correct in our system, since the satellite signals come from three different repeaters. This result in erroneous the off-the-shelf receiver position.

The data of each satellite, received by the off-the-shelf receiver on the up-converting receiver, are recorded. The recorded data include GPS satellite ID numbers, satellite azimuth and elevation angles, carrier-to-noise ratios (CNR) related to each satellite signal, and the GPS time of the week. GPS satellites transmit ephemeris through which the receiver can estimate the position of the satellites in the Earth-centered Earth-fixed (ECEF) coordinates system. In addition to a GPS satellite’s location (current and predicted), ephemeris includes the orbital parameters, clock bias, date, timing, health, and an almanac (a reduced subset of ephemeris of all satellites) exist. The ephemeris data can be collected using an online server to enable hot start in the proposed system. Moreover, the ephemeris data are used to calculate the satellite’s clock bias.

Each of the collected pseudo-range is a sum of the following distance terms: satellite-to-repeater distance, the receiver clock bias, satellite clock bias, repeater delay, the indoor distance from the corresponding repeater to the indoor receiver position.

The second step of the MATLAB^®^ algorithm is designed for choosing the GPS satellites, which are seen from the repeaters. For each repeater, specific satellites are chosen based on their CNR levels and locations. Each satellite’s position, clock bias, and distance from the corresponding repeater are calculated using the satellite’s ephemeris data. In the third step of the algorithm, the pseudo-range (PR) from each repeater to the selected satellite for the corresponding repeater is cleaned by subtraction operation on the right-hand side of Equation (1). Thus, the indoor distance from the receiver to each repeater is calculated. The calculated indoor distances also include the receiver clock bias. The indoor distances between each repeater and the indoor receiver is calculated using Equation (1).
(1)$$d_{i}+t_{bias}^{receiver}=PR_{Ri}^{Sj}-\left(D_{Ri}^{Sj}+\tau_{i}\times c+t_{bias}^{Sj}\times c\right)$$

In Equation (1), i is the index of the repeaters (i  = 1, 2, 3), j is the index of the satellites (j  = 1, 2,…, n), Ri represents the i^th^ repeater, Sj represents the j^th^ satellite, $d_{i}$ represents the indoor distance from repeater Ri to the receiver (with a clock bias of $t_{bias}^{receiver}$), $D_{Ri}^{Sj}$ stands for the distance from ith repeater (Ri) to the j^th^ satellite (Sj), $\tau_{i}$ is the propagation delay of the ith repeater, $t_{bias}^{Sj}$ is the satellite clock bias of the jth satellite, and c represents the speed of the light. When more than one satellite is chosen for a repeater, the $d_{i}+t_{bias}^{receiver}$ value is calculated by averaging the calculations from each satellite selected for that repeater. Thus, all the distances in Figure 2 can be solved with the proposed system.

The final step of the proposed algorithm is using the LSNAV algorithm to calculate the 2D indoor position for each sample collected. The indoor distances between the receiver and each repeater with receiver clock bias are calculated using Equation (1). Then, these calculated values and the repeater positions are given as the inputs to the LSNAV algorithm. It is important to note that the calculated indoor distances still have a receiver clock bias. Since the receiver clock bias ($t_{bias}^{receiver}$) is the same in all indoor distances, it is removed by the LSNAV algorithm by subtracting them from each other, and the indoor position is calculated. This algorithm is run for each sample. The obtained results are filtered with a moving average filter, and the indoor position is estimated as an average of these filtered points. More details regarding the estimation are provided under Section 4.

## 3. Experimental Setups for Indoor Positioning

The indoor positioning experiments were performed in the Sabancı University Faculty of Engineering and Natural Sciences. The floorplan is depicted in Figure 8.

The satellite positions in the sky have been observed as depicted in Figure 9, and the directional GPS antenna orientations are set accordingly.

The hardware delays from GPS antennas, amplifiers, attenuators, filters, converters, bias-tees, and cables are measured and presented as group delays ($\tau_{i}$) in Table 3. The group delay of each repeater is measured with a 20 GHz oscilloscope. As a result, for each repeater, GPS antenna and cabling have different measured values in the order of nanoseconds. These values are used in Equation (1) to calibrate the system. Moreover, each repeater position is also provided in Table 3. As mentioned earlier, two experiments are presented in this paper. While the repeater positions are fixed in the same location in both experiments, as presented in Table 3, the receiver position is different for each experiment. In the first experiment, the receiver is located at the intersection of 2 corridors, in that it is on the line-of-sight of all 3 repeaters. However, in the second experiment, the receiver is located closer to Repeater 1 (R1), in that it is not on the line-of-sight of Repeater 2 (R2).

In all experiments, there has been no switching, and all three repeaters have been transmitted simultaneously.

### 3.1. Setup for Experiment 1

Figure 10 shows the first experimental setup and indoor distances between each receiver and repeater. The repeater configurations are kept as provided in Table 3. Although Repeater 2 has a third amplifier, its attenuator is set to 15 dB. Therefore, its gain is 7 dB higher from Repeaters 2 and 3. The outdoor GPS antenna directions are set such that the beam of an antenna does not overlap in azimuth with other GPS antennas in the setup. The directional GPS antennas in Repeaters 1, 2, and 3 are, respectively, towards the geographic east, south, and west. In this experiment, the receiver is located at the intersection of two perpendicular corridors to provide a line-of-sight condition.

The real values of the indoor distances $d_{1}$, $d_{2}$, and $d_{3}$ are determined with a physical measurement using a laser pointer for a later comparison with the estimated values. The distance measurements are performed with respect to the known coordinates of the building.

### 3.2. Setup for Experiment 2

Figure 11 shows the second experimental setup and indoor distances between each receiver and repeater. In this experiment, a non-line-of-sight condition is formed by locating the receiver closer to Repeater 1. Repeater 2 does not directly see the receiver due to the corner at the intersection of two perpendicular corridors. The repeater configurations are kept as provided in Table 3. Repeater 2, which has an additional amplifier, is used with 5 dB attenuation to compensate for the non-line-of-sight condition and the scattering from the corner. Therefore, its gain is 17 dB higher than Repeaters 2 and 3.

The outdoor GPS antenna directions are the same as Experiment 1.

The real values of indoor distances $d_{1}$, $d_{2}$, and $d_{3}$ are determined with a physical measurement using a laser pointer for a later comparison with the estimated values. The distance measurements are performed with respect to the well-known coordinates of the building.

## 4. Results and Discussion

In this section, the techniques performed for the indoor position estimation with the proposed hardware, algorithm, and methods are presented along with the experimental data gathered in the experiments in a real indoor environment. Among these, the estimated indoor position, the satellites seen by each directional outdoor GPS antenna during the experiments, estimated distances from each repeater to the receiver with receiver clock bias, the CNR of each satellite signal within the angle of view of the corresponding repeater, and the 50% CEP from the estimated position are graphically visualized and presented in this section. The MATLAB^®^ routine with LSNAV algorithm, presented in Section 2, has been utilized for the position estimation of each sample collected in the experiments.

The satellites, in the angle of view of each repeater in Experiments 1 and 2, are presented in Figure 12. It should be noted that for this approach to work properly, it is important to know which repeater transfers the signal, coming from a particular satellite, into the building. This is in fact one of the reasons for the decision to utilize directional GPS antennas.

The association of the repeater, used for the signal of a particular GPS satellite, is accomplished by using the position, the azimuth, and the elevation of the directional GPS antenna and considering the GPS satellites that are in the targeted region of that particular antenna. If a satellite falls into the targeted region of multiple GPS antennas, we do not use any measurement based on that satellite, since we cannot be sure which repeater transferred the signal of that satellite into the building. In these experiments, therefore, the satellite labeled as G19 in Figure 12 (red point) has been ignored, while the satellites represented with green points have been utilized in position estimation for both experiments. Additionally, although the directional antenna beam widths are 60 degrees, the experiment results have shown that the GPS antennas can receive signals from a 90-degree angle in azimuth (Figure 12).

### 4.1. Results of Experiment 1

The first experiment is performed in the real indoor environment, which is presented in Figure 10, to demonstrate the performance of the proposed indoor positioning system under line-of-sight conditions. The position for each sample is estimated with the LSNAV algorithm using the GPS satellite’s signal, which is within the angle of view of each repeater. As mentioned previously, G19 is ignored because it is seen by more than one repeater. Under these conditions, the CNR of the received GPS signals, from the selected satellites for the indoor position estimation, is provided in Figure 13.

The sum of the indoor distances ($d_{1}$, $d_{2}$, and $d_{3}$), including receiver clock bias ($t_{bias}^{receiver}$), are calculated in Equation (1) and subtracted from each to obtain the difference terms ($d_{2}$ + $\;t_{bias}^{receiver}$)−($d_{1}$ + $t_{bias}^{receiver}$) and ($d_{3}$ + $t_{bias}^{receiver}$)−($d_{1}$ + $t_{bias}^{receiver}$) for each sample. Figure 14a,b shows the indoor distance difference terms in which the receiver clock biases are removed due to the subtraction, while in Figure 14c, it demonstrates the individual distances with the receiver clock bias ($d_{1}$ + $t_{bias}^{receiver}$, $d_{2}$ + $t_{bias}^{receiver}$, $d_{3}$ + $t_{bias}^{receiver}$).

As seen in Figure 14c, the indoor distances with receiver clock bias ($d_{1}$ + $t_{bias}^{receiver}$, $d_{2}$ + $t_{bias}^{receiver}$, $d_{3}$ + $t_{bias}^{receiver}$) can go up to thousands of meters. When the receiver clock bias is removed, with subtraction, the indoor differences can be found in the range of tens of meters. The calculated indoor distance difference terms (blue curve in Figure 14a,b) are averaged with a moving average filter of a five-sample window size. The resulting terms (red curve in Figure 14a,b) show a closer result to the measured indoor distance differences (black dashed line in Figure 14a,b), which can be calculated using the measured indoor distances $d_{1}$, $d_{2}$, and $d_{3}$ in Figure 10.

Figure 15a depicts the GPS repeater and outdoor GPS antenna positions according to their latitude and longitude in Experiment 1. These results show that the LSNAV algorithm has resulted in a solution for 74 out of 100 collected samples. The reason why the LSNAV does not result in a solution for every sample collected is addressed subsequently in the CNR provided in Figure 13. For some samples, due to environmental changes or multipath, an abrupt change in CNR occurs. For these samples, that do not have good CNR from all three repeaters, LSNAV does not result in a solution.

Moreover, a moving average filter with a five-sample window size has been applied over the 74 results from LSNAV, which then give 69 location estimations. The resulting 69 locations are depicted in Figure 15a,b with red dots. Figure 15a presents the results on a latitude–longitude graph, whereas Figure 15b presents the results on a local reference frame where the center is the real position of the receiver, which was previously determined by physical measurements in the experimental setup with respect to the building whose coordinates and plan are well known. We calculated the radius of 50% CEP as 3.3 m and plotted it in Figure 15a. Moreover, the indoor position is estimated as the mean of these 69 points obtained with moving average filtering. The real position of the receiver is represented with a black square, while the estimated position is presented with a blue square in Figure 15a,b. The estimated location is obtained by averaging 69 points, which are the output of the moving average filter, and is 54 cm away from the real position, as can be seen in Figure 15b.

Because the estimated position from 69 points is 54 cm away from the real position (Figure 15), a sub-meter accuracy has been achieved for line-of-sight conditions. In addition to the 50% CEP with a radius of 3.3 m from the estimated position, we also calculated each of the 69 points distances from the real position and presented it in Figure 16. The average error of the individual samples, from the real position, is 3.37 m.

### 4.2. Results of Experiment 2

Experiment 2 setup in Figure 11 is set to demonstrate the performance of the proposed indoor positioning system under non-line-of-sight conditions in a real indoor environment. The CNR of GPS signals received from the selected GPS satellites are provided in Figure 17.

Similar to Experiment 1, the average position is estimated with the LSNAV algorithm. In this experiment also, G19 is ignored, as it is within the angle of view of more than one repeater. Additionally, although G15 is seen by Repeater 2, its CNR is much lower than G24; therefore, it is also ignored in the calculations.

The indoor distances ($d_{1}$, $d_{2}$, and $d_{3}$), with receiver clock bias ($t_{bias}^{receiver}$) and their difference at each sample ($d_{2}$ + $t_{bias}^{receiver}$)−($d_{1}$ + $t_{bias}^{receiver}$) and ($d_{3}$ + $t_{bias}^{receiver}$)−($d_{1}$ + $t_{bias}^{receiver}$) are plotted and demonstrated in Figure 18. The resulting terms (red curve in Figure 18a,b) show a closer result to the measured indoor distance differences (black dashed line in Figure 18a,b), which can be calculated using the measured indoor distances $d_{1}$, $d_{2}$, and $d_{3}$ in Figure 11.

Figure 19a depicts the GPS repeater and outdoor GPS antenna positions, according to their latitude and longitude, in Experiment 2. In this experiment, the LSNAV algorithm has resulted in solutions for 35 out of 100 collected samples. Then, a moving average filter with a five-sample window size has been applied over these 35 converged solutions, and 30 location estimations have been obtained (Figure 19a,b with red dots). Figure 19a demonstrates the results on a latitude–longitude graph, whereas Figure 19b presents the results on a local reference frame in which the center is the real position of the receiver, which was previously determined by distance measurement in the experimental setup with respect to the known coordinates of the building structure. The indoor position is estimated as the mean of these 30 points. The real position of the receiver (black square) and estimated position (blue square) have been plotted in Figure 19a,b. It is seen that the estimated position is 98 cm away from the real position in Figure 19b.

The 50% CEP is calculated to have a radius of 4 m and drawn also in Figure 19a.

Moreover, the distance error from the real receiver position for each of the 30 points has been depicted in Figure 20. The average error of the individual samples from the real position is 4.17 m.

Under non-line-of-sight conditions in the experiments, while estimated position error increases, it is still possible to achieve sub-meter accuracy. To quantify the increased error amount, one can compare the radius of the 50% CEP circles in Figure 15a and Figure 19a. The radius of the 50% CEP is higher in Experiment 2. Additionally, when CNR values in Figure 17 are compared with those in Figure 13, we can conclude that the non-line-of-sight condition reduces the CNR. In the proposed system, variable attenuators are used to compensate for the non-line-of-sight conditions. To adjust, we can easily set the attenuation level, the retransmitted GPS power levels, and adjust the CNR levels so that the receivers can receive almost equal signal levels from each direction and each satellite. That is the reason why the attenuation level of Repeater 2 is set to a lower level in Experiment 2.

## 5. Conclusions

The implemented 433 MHz down-converting repeaters with variable attenuators and 433 MHz up-converting receiver with variable attenuator have been used in 2D positioning successfully for the first time. It is also shown that, with the help of cascaded LNA and variable attenuator, the proposed system for 2D positioning has gained immunity to the non-line-of-sight conditions. The power level of the signals that are transmitted to the receiver can be adjusted by varying the attenuation so that a receiver can pick up different satellite signals coming from different indoor paths almost equally. The GPS signals, which are down-converted and retransmitted by repeaters, are sent to the indoor environment and picked-up by an up-converting receiver. After up-conversion, the raw data are collected by an off-the-shelf receiver and then transmitted over Wi-Fi to a remote computer for processing. The estimated positions are found to be only 54 and 98 cm away from the actual receiver position, for line-of-sight and non-line-of-sight cases, respectively. Therefore, experiment results show that sub-meter accuracy can be achieved with the transmission of GPS signals in the 433 MHz ISM band in an indoor environment.

## Figures and Tables

**Figure 1 sensors-21-04338-f001:**
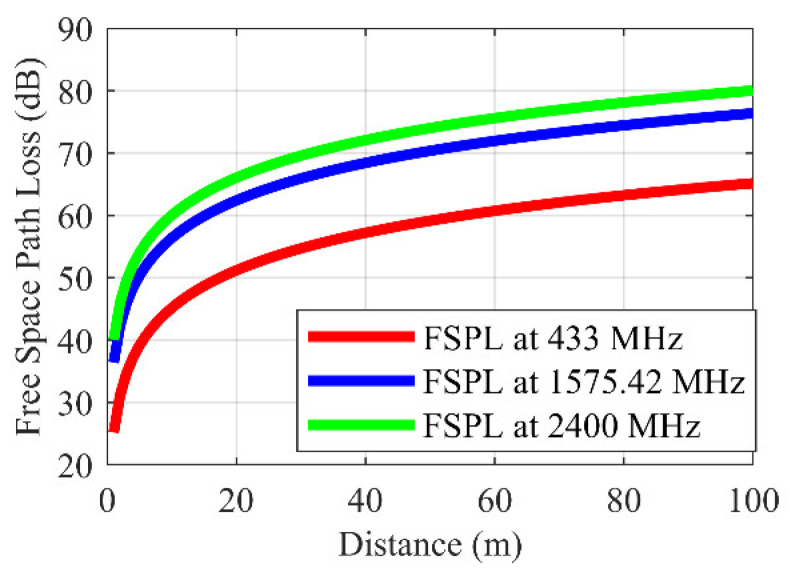
Free space path loss at 2.4 GHz, 1575.42 MHz, and 433 MHz.

**Figure 2 sensors-21-04338-f002:**
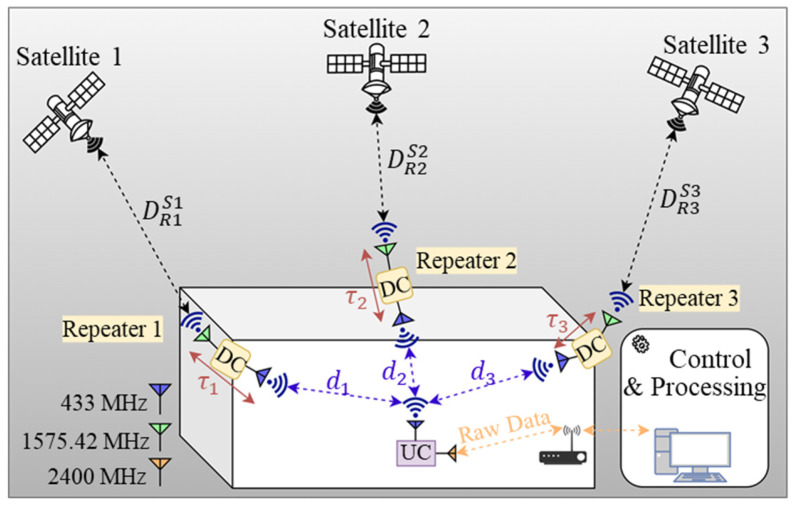
The proposed indoor positioning system. i*:* index of the repeaters (i = 1, 2, 3); j: index of the satellites (j  = 1, 2, …, n ); Ri: ith repeater; Sj: jth satellite; $d_{i}$: indoor distance from repeater Ri to the receiver (with a clock bias of $t_{bias}^{receiver}$ ); $D_{Ri}^{Sj}$: distance from ith repeater (Ri:) to the jth satellite (Sj ); $\tau_{i}$: propagation delay of the *i*th repeater; $t_{bias}^{Sj}$: satellite clock bias of the jth satellite.

**Figure 3 sensors-21-04338-f003:**
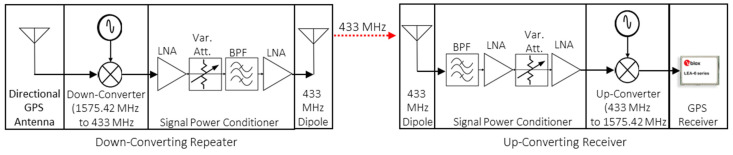
RF blocks in down-converting repeater (**left**) and up-converting receiver (**right**).

**Figure 4 sensors-21-04338-f004:**
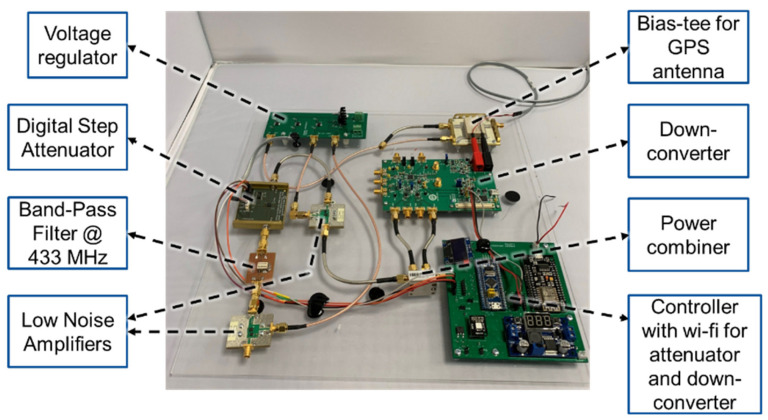
Implemented GPS down-converting repeater subsystem excluding the antennas.

**Figure 5 sensors-21-04338-f005:**
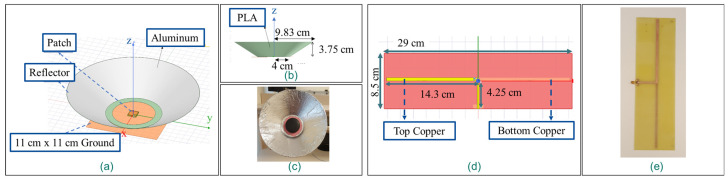
Modelled directional GPS antenna (**a**,**b**), fabricated directional GPS antenna (**c**), modelled 433 MHz dipole antenna (**d**), fabricated 433 MHz dipole antenna (**e**).

**Figure 6 sensors-21-04338-f006:**
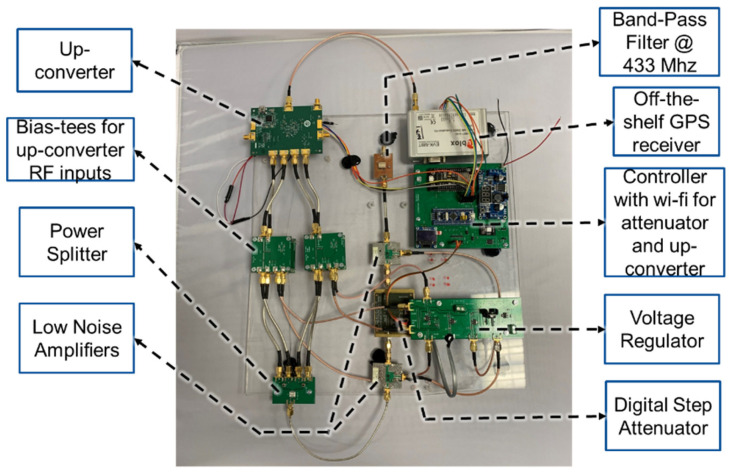
Implemented up-converting receiver excluding the antenna.

**Figure 7 sensors-21-04338-f007:**
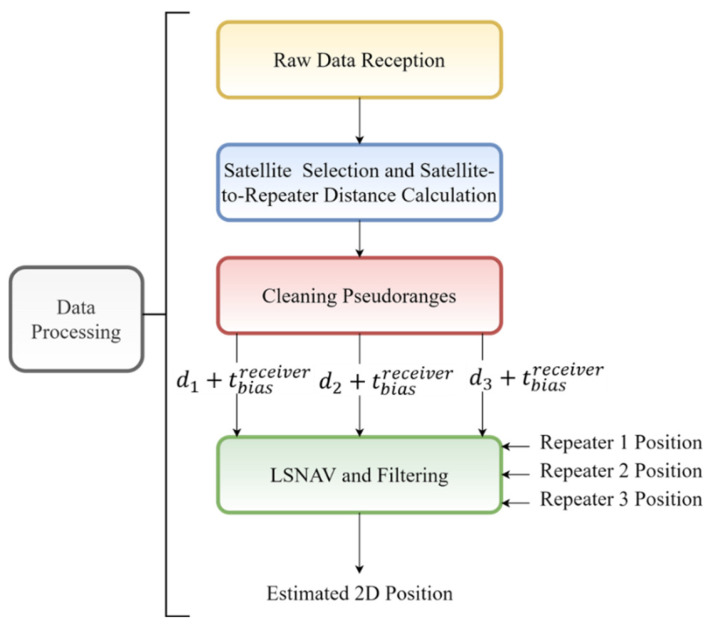
Flow of the algorithm.

**Figure 8 sensors-21-04338-f008:**
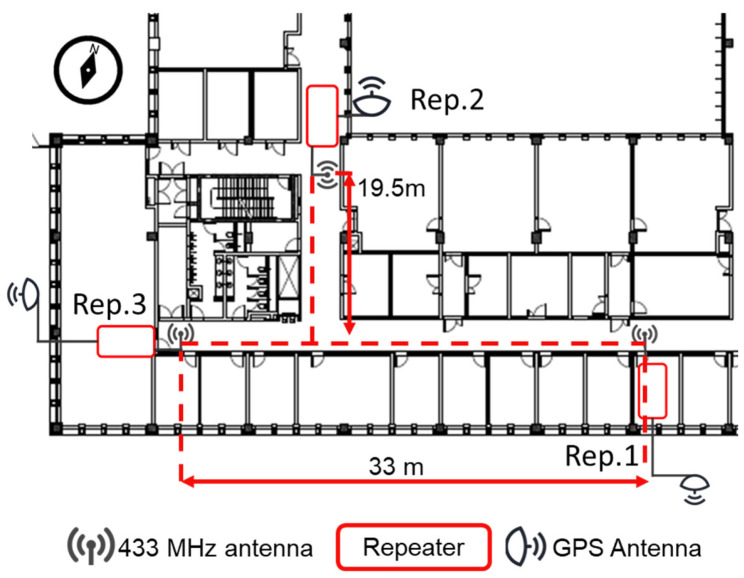
The location of repeaters on the floor plan of Sabancı University Faculty of Engineering and Natural Sciences (FENS) Building 2nd Floor.

**Figure 9 sensors-21-04338-f009:**
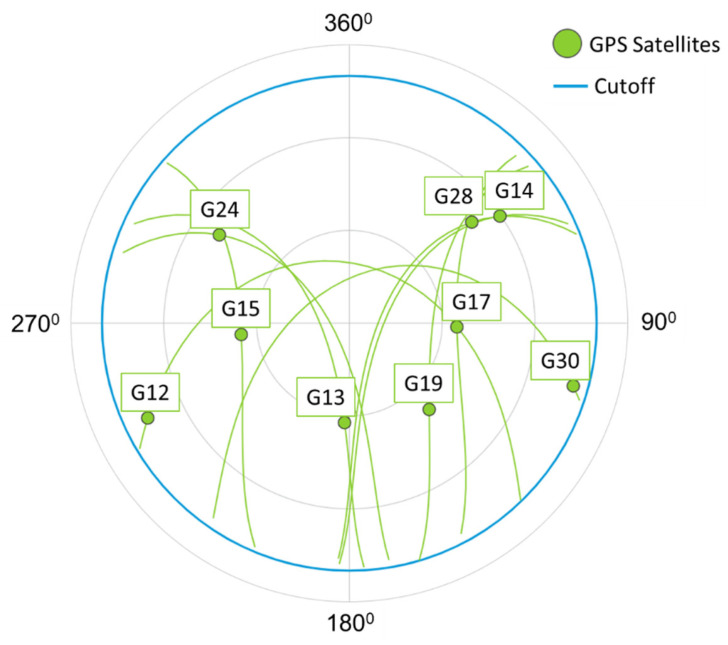
The satellite locations in the sky during the experiments.

**Figure 10 sensors-21-04338-f010:**
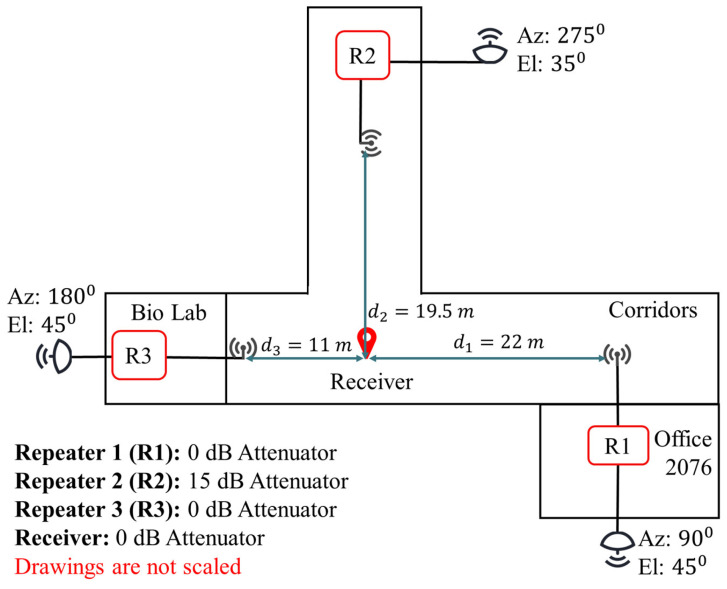
Distance between the receiver and each repeater, directional GPS antenna azimuth (Az) and elevation (El), and the attenuator value in each repeater in Experiment 1.

**Figure 11 sensors-21-04338-f011:**
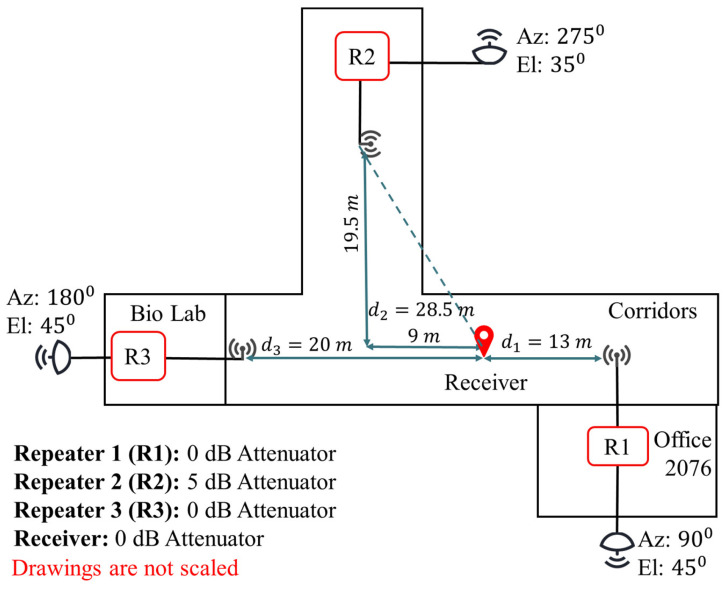
Distance between the receiver and each repeater, directional GPS antenna azimuth (Az) and elevation (El), and the attenuator value in each repeater in Experiment 2.

**Figure 12 sensors-21-04338-f012:**
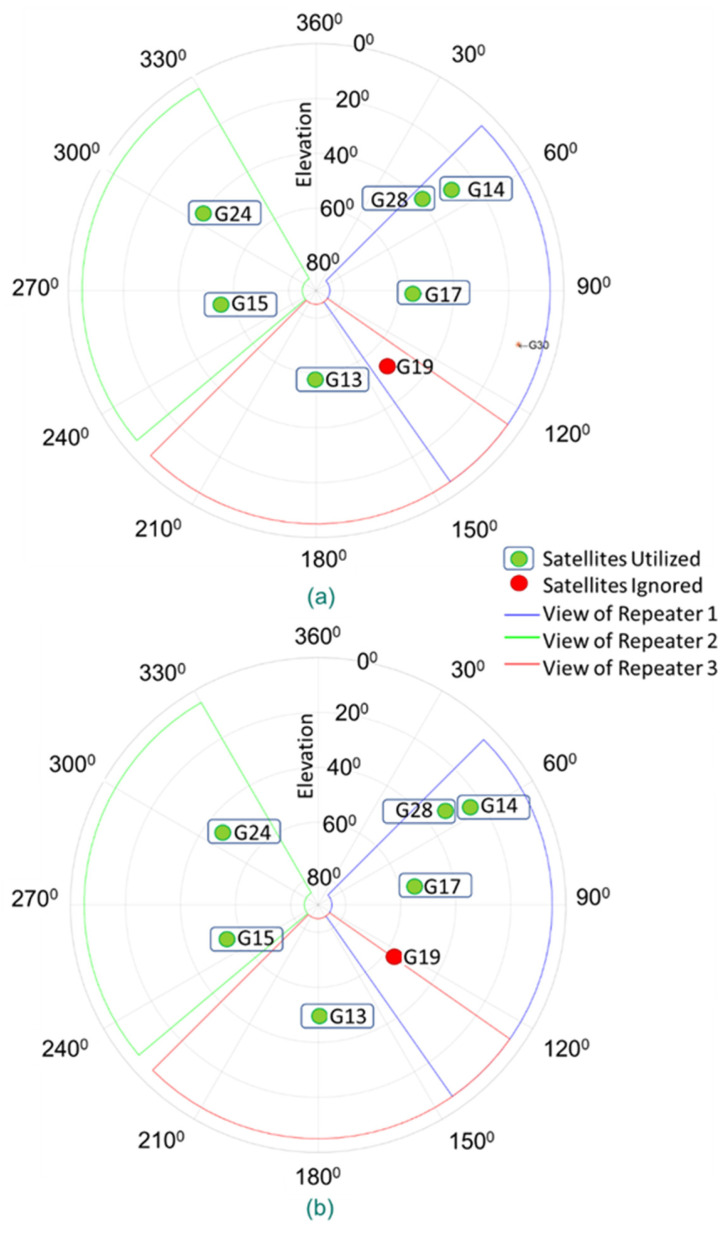
Satellites in each repeater’s angle of view in Experiments (**a**) 1 and (**b**) 2.

**Figure 13 sensors-21-04338-f013:**
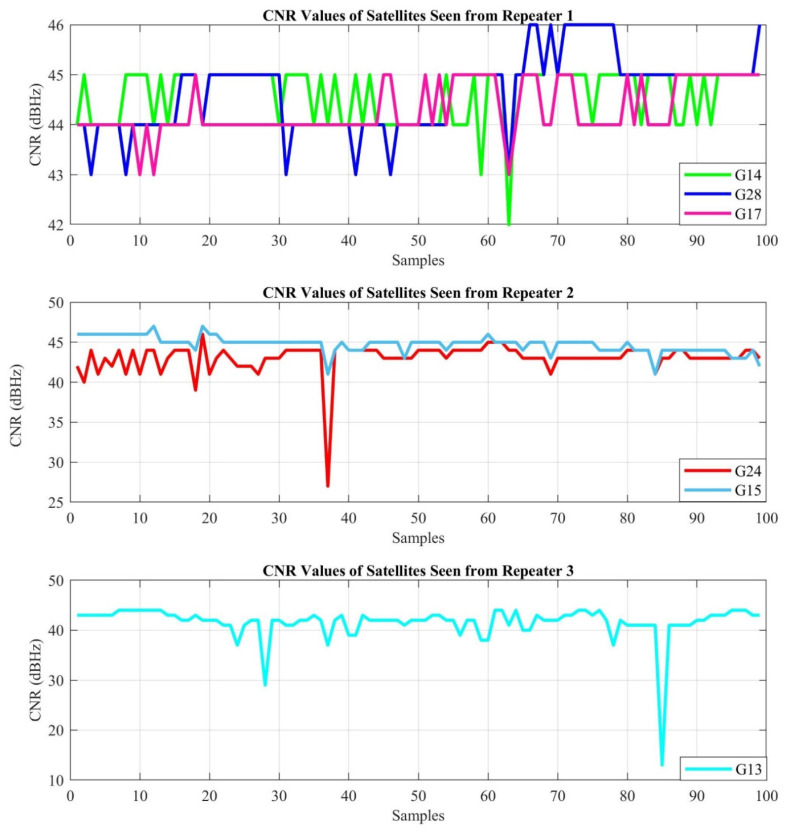
CNR of each GPS signal received by the repeaters.

**Figure 14 sensors-21-04338-f014:**
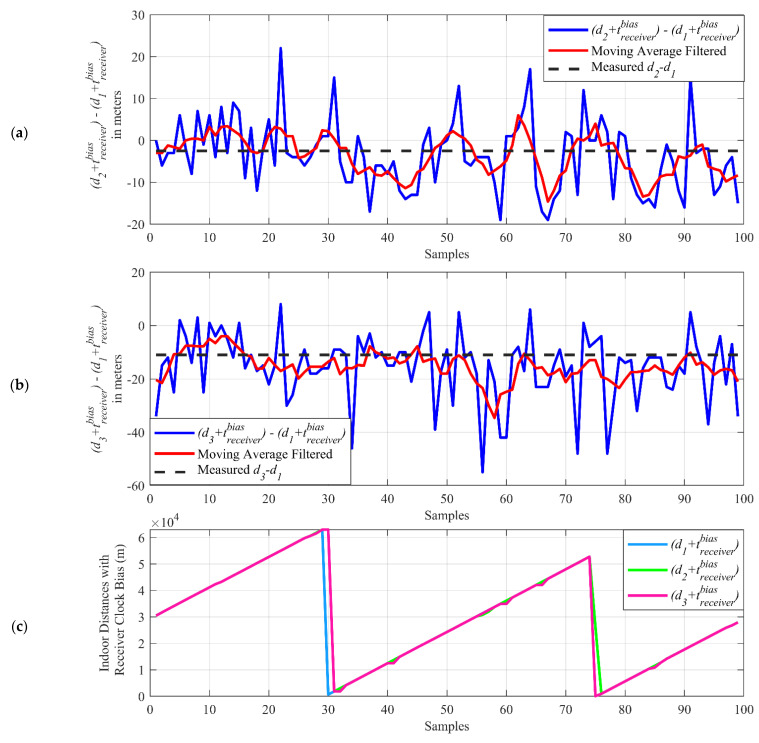
The indoor distance differences (**a**,**b**) and indoor distances with the receiver clock bias term (**c**).

**Figure 15 sensors-21-04338-f015:**
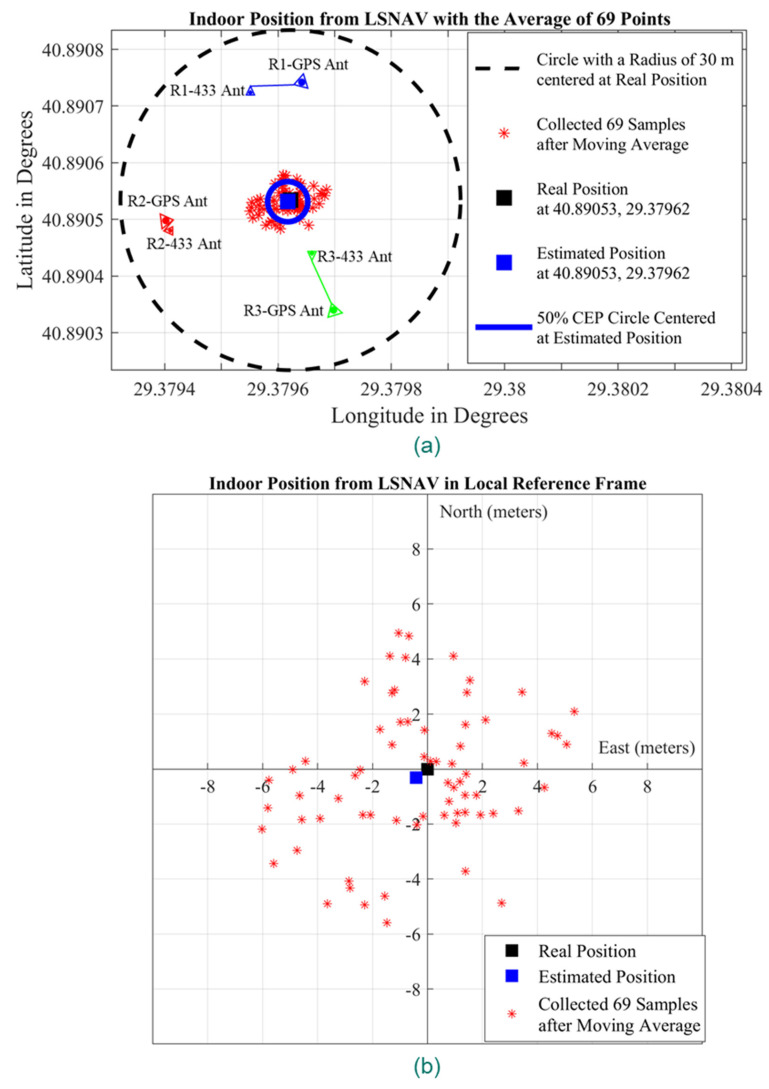
The results of the Experiment 1: (**a**) repeater positions, outdoor GPS antenna positions, collected samples (red dots), real position of the receiver (black square), estimated position (blue square), and CEP (blue circle) on latitude–longitude graph; (**b**) collected samples (red dots), real position of the receiver (black square), estimated position (blue square) on local reference frame whose center is the real position of the receiver.

**Figure 16 sensors-21-04338-f016:**
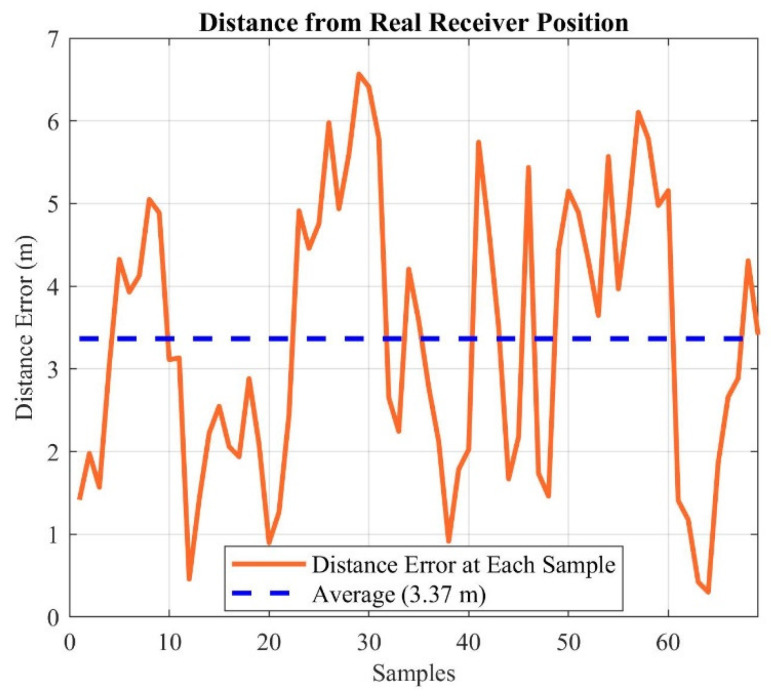
Distance from the real position for each of 69 points.

**Figure 17 sensors-21-04338-f017:**
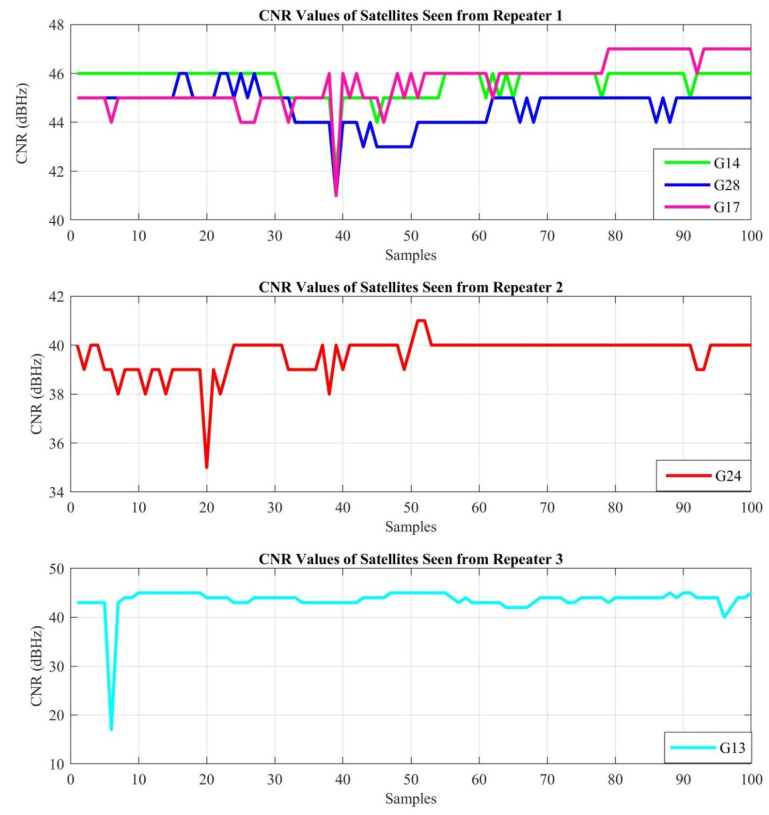
CNR of each GPS signal received by the repeaters.

**Figure 18 sensors-21-04338-f018:**
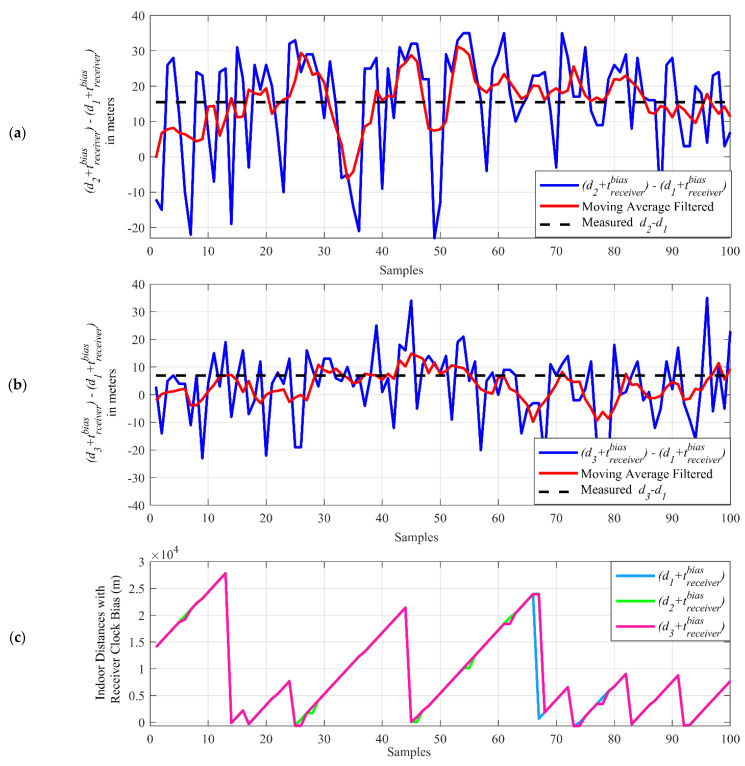
The indoor distance differences (**a**,**b**) and indoor distances with the receiver clock bias term (**c**).

**Figure 19 sensors-21-04338-f019:**
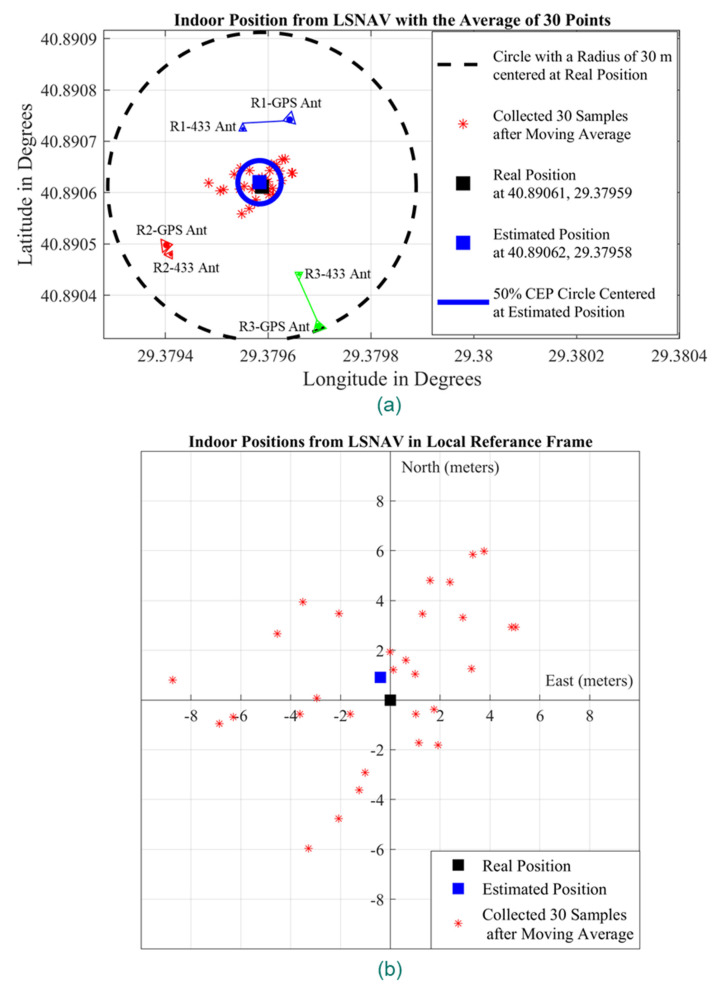
The results of the Experiment 2: (**a**) repeater positions, outdoor GPS antenna positions, collected samples (red dots), real position of the receiver (black square), estimated position (blue square), and CEP (blue circle) on latitude–longitude graph; (**b**) collected samples (red dots), real position of the receiver (black square), estimated position (blue square) on local reference frame whose center is the real position of the receiver.

**Figure 20 sensors-21-04338-f020:**
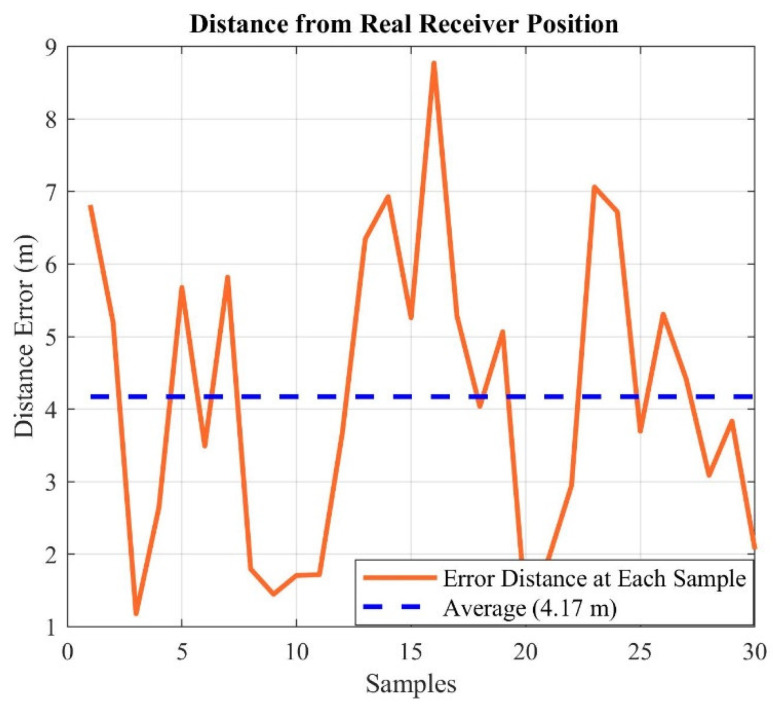
Deviation from the real position for each of 30 points.

**Table 1 sensors-21-04338-t001:** Repeater subsystem and its components.

Blocks	Components	Block Properties
Directional GPS Antenna	Tango 20 off-the-shelf active GPS antenna with conic reflector	34.45 dBi gain 60 degree 3 dB beam width
Directional GPS Antenna	TB-JEBT-4R2GW + bias tee	Provides DC to GPS antenna and transmits RF to down converter
Down-Converter	ADRF6820 quadrature demodulator,	8.7 dB block loss Programmable over SPI from controller block
Down-Converter	ZX10Q-2-5-S 90 degree power combiner	Combines down-converter I/Q outputs
Signal Power Conditioner	LHA-13LN + low noise amplifier	22.43 dB gain 0.9 dB noise figure
Signal Power Conditioner	DAT-31R5A-SP + digital step attenuator (variable attenuator)	0 to 31.5 dB adjustable attenuation by 0.5 dB step size and can be adjusted from controller
Signal Power Conditioner	DBP.433.T.A.30 band pass filter at 433 MHz	1.7 dB insertion loss at 433 MHz, 19 MHz 3 dB bandwidth
Signal Power Conditioner	LHA-13LN + as the second low noise amplifier	22.43 dB gain 0.9 dB noise figure
433 MHz Dipole	433 MHz dipole on FR4	2.1 dBi gain 80 degree 3 dB beam width on both sides
Supporting Blocks	Voltage Regulator	Provides DC to system components and blocks
Supporting Blocks	Controller with Wi-Fi for programming attenuator and down-converter	Wi-Fi connection to remote PC, SPI connection to ADRF6820, and attenuator

**Table 2 sensors-21-04338-t002:** Receiver subsystem and its components.

Blocks	Components	Block Properties
433 MHz Dipole	433 MHz dipole on FR4	2.1 dBi gain, 80 degree 3 dB beam width on both sides
Signal Power Conditioner	DBP.433.T.A.30 band pass filter at 433 MHz	1.7 dB insertion loss at 433 MHz, 19 MHz 3 dB bandwidth
Signal Power Conditioner	LHA-13LN + low noise amplifier	22.43 dB gain 0.9 dB noise figure
Signal Power Conditioner	DAT-31R5A-SP + digital step attenuator (variable attenuator)	0 to 31.5 dB adjustable attenuation by 0.5 dB step size and can be adjusted from controller
Signal Power Conditioner	LHA-13LN + as the second low noise amplifier	22.43 dB gain 0.9 dB noise figure
Up-Converter	I/Q power divider, bias tees	Provides I/Q inputs and required DC levels to ADRF6720-27
Up-Converter	ADRF6720-27 quadrature modulator	1.2 dB block loss Programmable over SPI from controller block
Off-the-shelf GPS Receiver	LEA-6T chipset	
Supporting Blocks	Voltage Regulator	Provides DC to system components and blocks
Supporting Blocks	Controller with Wi-Fi for programming attenuator and down-converter	Wi-Fi connection to remote PC, SPI connection to ADRF6820, and attenuator

**Table 3 sensors-21-04338-t003:** Repeater configurations.

Repeater Number (Ri)	$\mathrm{Measured}\;\mathrm{Group}\;\mathrm{Delay}\;(\tau_{i})$	Position (Latitude, Longitude)
R1	48.5 ns	40.89072, 29.37955
R2	103 ns *	40.89048, 29.37941
R3	86 ns	40.89044, 29.37966

The latitudes and longitudes are given in degrees. * Repeater 2 has a third LHA-13LN+ amplifier.

## Data Availability

Not Applicable.

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
