# Peer review of "Indoor Positioning System Based on Global Positioning System Signals with Down- and Up-Converters in 433 MHz ISM Band"

_sensors, 2021, doi:10.3390/s21134338_

Round 1
Reviewer 1 Report
The paper presents a 2D indoor positioning system using GPS signals in 433 MHz ISM band. Its prototype was implemented and tested under line-of-sight and non-line-of-sight conditions, by utilizing down-converting repeaters and an up-converting one.
Some acronyms and abbreviations are not defined. They should be clearly defined or readers even though they are popular to the authors, because some readers might be not familiar with them. For example, GNSS and ISM are defined in Abstract.
The organization of the paper should be described in the last paragraph of Section 1 for readers.
In Section 2.3, more detailed presentation of the proposed algorithm should be given because Figure 7 and related explanations are not sufficient to catch the detailed operation of the algorithm. It is suggested to describe more clearly in detail, by giving additional figures or diagrams or pseudocodes.
The authors present only 2D indoor positioning system. Because 3D positioning system is ultimate goal in general indoor applications, it is highly recommended to discuss challenges and issues for 3D indoor positioning, in association with the authors’ results.
Author Response
Response to the Reviewers
The authors would like to thank the reviewers for their valuable constructive comments, which helped the authors in improving the technical content. The manuscript was carefully revised according to these comments.
Reviewer 1:
Comment #1: Some acronyms and abbreviations are not defined. They should be clearly defined or readers even though they are popular to the authors, because some readers might be not familiar with them. For example, GNSS and ISM are defined in Abstract.
Response to reviewer comment #1:
The authors thank the reviewer for the suggestion and comments. All abbreviations and acronyms were defined. The abbreviations in the abstract were added and the manuscript was modified accordingly:
“…in the 433 MHz Industrial Scientific Medical (ISM) band is..”
“..to avoid the restrictions on the use of Global Navigation Satellite Systems (GNSS) repeaters..”
Comment #2: The organization of the paper should be described in the last paragraph of Section 1 for readers.
Response to reviewer comment #2:
Further additions were made to the revised version of the manuscript to present the organization of the paper.
“The rest of the paper is organized as follows: Section 2 introduces the proposed indoor positioning system and describes its hardware, software and algorithms. Section 3 describes the experimental framework and the real-life environment where we tested our system. Section 4 analyzes and discusses the results we achieved when the proposed system and technique are used for 2D indoor positioning. Finally, Section 5 concludes this paper.”
Comment #3: In Section 2.3, more detailed presentation of the proposed algorithm should be given because Figure 7 and related explanations are not sufficient to catch the detailed operation of the algorithm. It is suggested to describe more clearly in detail, by giving additional figures or diagrams or pseudocodes.
Response to reviewer comment #3:
The content of Section 2.3 was enriched with more explanations. The algorithms developed and used were explained with more details and Figure 7 was modified in the revised version of the manuscript accordingly:
“In an indoor environment, although the signal loss and GPS coverage problems can be overcome with the proposed repeaters, this solution requires additional algorithms that takes the non-line of sight propagation and repeater delay into account. The proposed technique introduces a new path in that the distance between satellite and the receiver becomes different from that of the normal operation of an off-the-shelf receiver during which there is a line-of-sight distance between the satellite and the receiver. The GPS signals in the proposed scheme come to the repeater first, and then reach to the receiver as seen in Figure 2. “
“The raw data (such as pseudo-range, carrier-to-noise ratio, satellite azimuth and elevation angles, etc.) obtained from the off-the-shelf receiver is transmitted to the remote computer through the Wi-Fi connection.”
“The routine summarized in Figure 7 has the following four steps: raw data reception, satellite selection and satellite to repeater distance calculations, cleaning pseudorange from satellite-to-repeater distances and satellite biases, and finally running the Least Squares Navigation (LSNAV) algorithm, which is a least squares solution which minimizes the sum of square of the residual errors [48].”
“In the first part, we remove the biases from the GPS pseudo-range measurement using the models of the troposphere, ionosphere, GPS satellite clocks, GPS satellite movement during signal propagation, and Earth rotation.”
“Navigation data is also collected as part of raw data from the off-the-shelf receiver, which contains GPS time of the week (ITOW) of the navigation epoch. ITOW field indicates the GPS time at which the navigation epoch occurred. Each navigation solution is triggered by the tick of the 1 kHz clock nearest to the desired navigation solution time. This tick is referred to as a navigation epoch. If the navigation solution attempt is successful, one of the results is an accurate measurement of time in the time-base of the chosen GNSS system, called GNSS system time. The difference between the calculated GNSS system time and receiver local time is called the clock bias (and the clock drift is the rate at which this bias is changing). Navigation data also contains Earth Centered Earth Fixed (ECEF) coordinates solution of the off-the-shelf receiver, however, this solution would not be correct in our system since the satellite signals come from 3 different repeaters. This result in erroneous the off-the-shelf receiver position.”
“GPS satellites transmit ephemeris through which the receiver can estimate the position of the satellites in the Earth Centered Earth Fixed (ECEF) coordinates system. In addition to a GPS satellite’s location (current and predicted), ephemeris includes the orbital parameters, clock bias, date, timing, health, and an almanac (a reduced subset of ephemeris of all satellites) exist. The ephemeris data can be collected using an online server to enable hot start in the proposed system.”
“Each of the collected pseudorange is a sum of the following distance terms: satellite to repeater distance, the receiver clock bias, satellite clock bias, repeater delay, the indoor distance from the corresponding repeater to the indoor receiver position.”
“The second step of the MATLAB® algorithm is designed for choosing the GPS satellites which are seen from the repeaters.”
“... the pseudorange (PR) from each repeater to the selected satellite for the corresponding repeater is cleaned by subtraction operation on the right-hand side of Equation 1. Thus, the indoor distance from the receiver to each repeater is calculated.”
Figure 7. Flow of the Algorithm
Comment #4: The authors present only 2D indoor positioning system. Because 3D positioning system is ultimate goal in general indoor applications, it is highly recommended to discuss challenges and issues for 3D indoor positioning, in association with the authors’ results.
Response to reviewer comment #4:
The authors thank for the reviewer's suggestion. Suggested indoor positioning system can be used for 3D positioning as well with an additional 4th repeater. 3D positioning is not studied within the scope of this study in which the aim is to determine the position of an object located on a single floor. The following explanation on 3D positioning with the proposed system was added to the revised version of the manuscript to the readers:
“In this particular paper, we present two experiments for 2D indoor positioning in the 433 MHz ISM band: one experiment is under line-of-sight conditions, and another is under non-line-of-sight conditions. Note that adding a 4th repeater will allow us to achieve 3D indoor positioning.”

Reviewer 2 Report
The paper proposes an indoor positioning system based on GPS signals relayed in the 433 MHz ISM frequency band through a down- and up-converting scheme.
The paper shows an interesting technique for using GPS signals in indoor environment. A complete set of hardware and software components have been developed and tested in a real environment. The presentation and the technical description in the first part of the paper (until Section 2.2 comprised) is good, whereas the second part of the work must be improved. Below, some major and minor comments the author should address:
- Section Algorithm for Indoor Position Estimation is quite confusing in many parts.
- “The raw data obtained from the off-the-shelf receiver is transmitted”. Please clarify what the author means by “raw data”;
- “GPS satellites also transmit their location (current and predicted)”. Please clarify this claim: GPS satellites transmit ephemeris through which the receiver can estimate the position of the satellites;
- “the pseudorange (PR) … is refined and the indoor distances from the receiver to the repeaters are calculated.”.Please rephrase this sentence.
- The clock bias terms in eq.1 are not easy to read. Please change the symbol representation.
- Please provide some more details about the LSNAV algorithm used.
- Please add a reference about peer-to-peer cooperative positioning for indoor positioning (e.g. “Peer‐to‐peer cooperation for GPS positioning”);
- What is the position solution rate?
- Fig.14: there are typos in the legend; moreover, some concerns about the filtering operation: i) using moving average filtering in the computed indoor distances will introduce a processing delay; ii) the filtering operation in the distance information before applying the LSNAV algorithm could provide a not reliable position solution; finally, it is not clear what actually is the “Actual difference”;
- Fig.15: Please provide the results in the local reference frame such as north-east frame (not latitude and longitude); the legend is not readable in this form;
- Fig.16: Distance error and not “Error Distance”; how did the author determine the real position of the receiver? “Distance error” should be in the y-axis and not in the legend (the explanation that the distance error refers to the difference between the measurement and the ground truth should be included in the figure caption); the same comments for Fig. 18 and Fig.19.
Author Response
Response to the Reviewers
The authors would like to thank the reviewers for their valuable constructive comments, which helped the authors in improving the technical content. The manuscript was carefully revised according to these comments.
Reviewer 2:
Comment #1: Section Algorithm for Indoor Position Estimation is quite confusing in many parts.
Response to reviewer comment #1:
The authors thank the reviewer for the comment. Section Algorithm for Indoor Position Estimation was explained with further details to increase the clarity of the section.
Comment #2: “The raw data obtained from the off-the-shelf receiver is transmitted”. Please clarify what the author means by “raw data”;
Response to reviewer comment #2:
The term raw data in Section 2.3 was explained with further details in the revised version of the manuscript, and manuscript was modified accordingly:
“The raw data (such as pseudo-range, carrier-to-noise ratio, satellite azimuth and elevation angles, etc.) obtained from the off-the-shelf receiver is transmitted to the remote computer through the Wi-Fi connection.”
Comment #3: “GPS satellites also transmit their location (current and predicted)”. Please clarify this claim: GPS satellites transmit ephemeris through which the receiver can estimate the position of the satellites;
Response to reviewer comment #3:
The revised version of the manuscript was modified accordingly:
“GPS satellites transmit ephemeris through which the receiver can estimate the position of the satellites in the Earth Centered Earth Fixed (ECEF) coordinates system. In addition to a GPS satellite’s location (current and predicted), ephemeris includes the orbital parameters, clock bias, date, timing, health, and an almanac (a reduced subset of ephemeris of all satellites) exist. The ephemeris data can be collected using an online server to enable hot start in the proposed system.”
Comment #4: “the pseudorange (PR) … is refined and the indoor distances from the receiver to the repeaters are calculated.”.Please rephrase this sentence.
Response to reviewer comment #4:
The authors thank for the comment. The term “refinement” was changed with the term “cleaning”. To further explain what we mean by “cleaning pseudorange” we added the following explanations and modified the manuscript accordingly:
“In the third step of the algorithm, the pseudorange (PR) from each repeater to the selected satellite for the corresponding repeater is cleaned by subtraction operation on the right-hand side of Equation 1. Thus, the indoor distance from the receiver to each repeater is calculated.”
Comment #5: The clock bias terms in eq.1 are not easy to read. Please change the symbol representation.
Response to reviewer comment #5:
Equation 1 is modified as suggested by the reviewer to increase the clarity in the manuscript accordingly:
“The indoor distances between each repeater and the indoor receiver is calculated using Equation 1.
|
(1) |
In Equation 1, is the index of the repeaters (=1, 2, 3), is the index of the satellites (=1, 2,…, ), represents the th repeater, represents the th satellite, represents the indoor distance from repeater to the receiver (with a clock bias of ), stands for the distance from ith repeater () to the th satellite (), is the propagation delay of the th repeater, is the satellite clock bias of the th satellite, and represents the speed of the light. When more than one satellite is chosen for a repeater, the value is calculated by averaging the calculations from each satellite selected for that repeater. Thus, all the distances in Figure 2 can be solved with the proposed system.”
Comment #6:
Please provide some more details about the LSNAV algorithm used.
Response to reviewer comment #6:
More explanation and a reference for the readers were provided on LSNAV algorithm in the revised version of the manuscript accordingly:
“..finally running the Least Squares Navigation (LSNAV) algorithm, which is a least squares solution which minimizes the sum of square of the residual errors [48].”
Comment #7:
Please add a reference about peer-to-peer cooperative positioning for indoor positioning (e.g. “Peer‐to‐peer cooperation for GPS positioning”);
Response to reviewer comment #7:
Peer-to-peer cooperative positioning for indoor positioning was added to the introduction in the revised version of the manuscript accordingly:
“... and peer-to-peer cooperative positioning [36].”
Comment #8:
What is the position solution rate?
Response to reviewer comment #8:
Currently we use u-Blox® (the off-the-shelf receiver) at 1 Hz rate and provide a solution at every 1.5 seconds. However, we can increase u-Blox rate to 2-5 Hz to obtain a solution under a second for time-critical applications.
Comment #9:
Fig.14: there are typos in the legend; moreover, some concerns about the filtering operation: i) using moving average filtering in the computed indoor distances will introduce a processing delay; ii) the filtering operation in the distance information before applying the LSNAV algorithm could provide a not reliable position solution; finally, it is not clear what actually is the “Actual difference”;
Response to reviewer comment #9:
The authors thank for the reviewer's comment. The typos in the legend of Figure 14 were correctedand modified accordingly:
Figure 14. The indoor distance differences (a and b) and indoor distances with the receiver clock bias term (c)
With filtering we aimed to remove the large spikes in the data. Regarding the process delay, we take first 5 samples to create the first solution and then at every next sample we generate a solution. Therefore the overhead is the first five sample’s arrival time. As mentioned for the previous comment, we can increase u-Blox rate to 2-5 Hz to obtain a solution under a second for time-critical applications. For the next concern, Filtering first and calculating position or calculating position first and then filtering could be explored for comparison between these two different approaches, however, it was not considered within the scope of this paper.
Comment #10:
finally, it is not clear what actually is the “Actual difference”;
Response to reviewer comment #10:
The legends in Figure 14 is modified and the term “actual difference” is changed by the term “measured d2-d1” and “Measured d3-d1”. These difference terms are explained in the revised version of the manuscript in Section 4 accordingly:
“The resulting terms (Red curve in Figure 14a and Figure 14b) show a closer result to the measured indoor distance differences (Black dashed line in Figure 14a and Figure 14b), which can be calculated using the measured indoor distances , , and in Figure 10.”
Moreover, in Section 3 we added the following explanation:
“The real values of indoor distances , , and are determined with a physical measurement using a laser pointer for a later comparison with the estimated values. The distance measurements are performed with respect to the known coordinates of the building. “
Comment #11:
Fig.15: Please provide the results in the local reference frame such as north-east frame (not latitude and longitude); the legend is not readable in this form;
The authors thank for the reviewer's suggestion. A new plot is added in Figure 15 (as Figure 15b). The revised version of the manuscript was modified accordingly:
“The resulting 69 locations are depicted in Figure 15a and b with red dots. Figure 15a presents the results on a latitude-longitude graph whereas Figure 15b presents the results on a local reference frame where the center is the real position of the receiver which was previously determined by physical measurements in the experimental setup with respect to the building whose coordinates and plan are well known. We calculated the radius of 50% CEP as 3.3 meters and plotted it in Figure 15a. Moreover, the indoor position is estimated as the mean of these 69 points obtained with moving average filtering. The real position of the receiver is represented with a black square while the estimated position presented with a blue square in in Figure 15a and b. The estimated location is obtained by averaging 69 points, which are the output of the moving average filter, is 54 cm away from the real position, as can be seen in Figure 15b.”
The modified Figure 15 is presented below:
Figure 15. The results of the Experiment 1: (a) Repeater positions, Outdoor GPS antenna positions, Collected Samples (red dots), real position of the receiver (black square), estimated position (blue square), and CEP (blue circle) on latitude-longitude graph; (b) Collected Samples (red dots), real position of the receiver (black square), estimated position (blue square) on local reference frame whose center is the real position of the receiver.
Comment #12: Fig.16: Distance error and not “Error Distance”; how did the author determine the real position of the receiver? “Distance error” should be in the y-axis and not in the legend (the explanation that the distance error refers to the difference between the measurement and the ground truth should be included in the figure caption);
Response to reviewer comment #12:
The author thanks for the comment. The term “distance error” is replaced with “error distance” and Figure 16 is modified in the revised version of the manuscript. To determine the real position of the receiver, we measured the distances of the repeaters with respect to the reference points in the building. We also determined the location of the receiver with respect to the building with a laser pointer. Then, we obtained the building map coordinates and coordinates of the reference points from the Google maps and relate them to our local measurements.To give an idea, we added the following to the revised version of the manuscript:
“The real values of the indoor distances , , and are determined with a physical measurement using a laser pointer for a later comparison with the estimated values. The distance measurements are performed with respect to the well-known coordinates of the building.”
Figure 16. Distance from the real position for each of 69 points.
Comment #13: the same comments for Fig. 18 and Fig.19.
Response to reviewer comment #12:
Authors thank to the reviewer for the comments. Figures 18, 19, and 20 are modified in line with comments 11 and 12.

Round 2
Reviewer 1 Report
The review comments have been responded appropriately and the paper has been revised accordingly.
Reviewer 2 Report
The author has properly addressed the reviewer's comments.